# Cystine and Theanine as Stress-Reducing Amino Acids—Perioperative Use for Early Recovery after Surgical Stress

**DOI:** 10.3390/nu14010129

**Published:** 2021-12-28

**Authors:** Takashi Tsuchiya, Shigekazu Kurihara

**Affiliations:** 1Department of Surgery, Sendai City Medical Center, Sendai 983-0824, Japan; tsuchiya@openhp.or.jp; 2Nutrition Care Group, Quick Nourishment Department, Food Products Division, Ajinomoto Co., Inc., Tokyo 104-8315, Japan

**Keywords:** cystine, theanine, glutathione (GSH), perioperative nutrition, surgical stress

## Abstract

Perioperative nutritional therapy requires the consideration of metabolic changes, and it is desirable to reduce stress aiming at early metabolic normalization. Glutathione (GSH) is a tripeptide composed of glutamic acid, cysteine, and glycine. It is one of the strongest antioxidants in the body and important for adjusting immune function. Cystine and theanine (γ-glutamylethylamide) provide substrates of GSH, cysteine and glutamic acid, promoting the synthesis of GSH. It has been reported that the ingestion of cystine (700 mg) and theanine (280 mg) exhibits inhibitory effects against excess inflammation after strong exercise loads in athletes, based on which its application for invasive surgery has been tried. In patients undergoing gastrectomy, ingestion of cystine (700 mg) and theanine (280 mg) for 10 days from 5 days before surgery inhibited a postoperative increase in resting energy expenditure, promoted recovery from changes in interleukin-6, C-reactive protein, lymphocyte ratio, and granulocyte ratio and inhibited an increase in body temperature. In a mouse small intestine manipulation model, preoperative 5-day administration of cystine/theanine inhibited a postoperative decrease in GSH in the small intestine and promoted recovery from a decrease in behavior quantity. Based on the above, cystine/theanine reduces surgical stress, being useful for perioperative management as stress-reducing amino acids.

## 1. Introduction

### Stress and Biological Reaction

When stress, such as surgery, infection, and burn, is loaded, the body exhibits a defense reaction and induces metabolic, immune, and endocrine changes. The changes in parameters simply monitored in routine clinical practice include fever, C-reactive protein (CRP) increase, peripheral blood leukocytosis, lymphocytopenia, and a decrease in urine volume. These inflammatory reactions become strong as stress increases and the time to recovery extends with it. Regarding the time-course metabolic changes, according to Moore et al., the stress period (phase 1) continues for 2–4 days, followed by a 4–7-day turning point (phase 2), and then the anabolic phase. Regeneration of muscle protein (muscle strength: phase 3) occurs one to several weeks after surgery, followed by the fat accumulation period (fat gain: phase 4), extending for several weeks to several months (Figure 1) [1]. Energy consumption increases due to stress. In the estimation of energy consumption using the Harris–Benedict equation, the coefficient multiplying the resting energy expenditure (REE) is set up to 1.0–2.0 in consideration of the severity of stress [2,3]. Under stress, synthesis of acute-phase proteins, such as CRP, ceruloplasmin, and fibrinogen, increases, and the synthesis of negative acute-phase proteins, such as albumin and transthyretin, decreases [4]. Muscle protein is decomposed and mobilized for tissue repair, gluconeogenesis, and energy production [5,6]. To estimate the degree of protein catabolism clinically, urinary urea nitrogen (UUN) is measured [7]. Normally, 7–13 g/day of UUN is excreted, but excretion increases in response to stress, and it may exceed 20 g/day in critically ill patients. Nitrogen balance is calculated from the nitrogen dose and nitrogen excretion, and the value becomes negative due to catabolic promotion under stress. After nutrition management, including protein and amino acid administration, the balance subsequently turns positive as the course enters the turning point [8,9]. Regarding nutrition management, nutrients administered after the protein anabolic phase are absorbed and utilized as usual, but a sufficiently effective utilization of nutrients exogenously administered in the stress period is impossible and excessive administration induces metabolic complications, thereby increasing stress [10]. In inpatient treatment aiming at surgery, treatment in the stress period over the turning point is performed and consideration of metabolic changes is necessary for perioperative management. In this review, the significance of nutritional therapy as a stress-reducing measure and the stress-reducing effects of the amino acids cystine and theanine, which promote glutathione (GSH) synthesis, are described.

## 2. Trial of Stress Reduction

### 2.1. Surgical Procedure

Surgery has both merits of treatment effects and demerits for the body, including surgical stress. Historically, it has been approached through both aspects: methods to extend surgery (resulting in overstress) to increase the treatment effects and to reduce stress. Laparoscopic surgery has recently been introduced, it was clarified that patients recover rapidly because surgery is completed without making a large laparotomy wound, and laparoscopic surgery is now recognized as a representative minimally-invasive surgery [11]. In a study in which the same surgical procedure was performed by laparotomy and laparoscopy and compared, analgesics were used less frequently in patients treated with laparoscopic surgery, and leukocytosis and increased CRP were inhibited, shortening the hospital stay [12,13,14]. Reduction of postoperative UUN excretion after laparoscopic surgery of the large intestine compared with that after laparotomic surgery was also observed in our department and shortening of hospital stay was achieved.

### 2.2. Nutritional Management (Early Enteral Feeding: Animal Model)

Mochizuki et al. investigated the influence of the difference in the timing of initiation of nutritional supplement administration on an increase in the energy consumption using a guinea pig model of a 30% body surface area burn [15]. In the group, in which enteral nutrition was initiated 72 h after injury, energy consumption increased to 148% of that before injury on day 9 after injury, but the increase remained at 112% in the group with early enteral nutrition initiated 2 h after injury, demonstrating a significant difference, i.e., the initiation of enteral nutrition management early after stress (surgery) reduces energy consumption, exhibiting stress-reducing effects as a modification of the surgical procedure, such as laparoscopic surgery, reduces stress, giving an important implication for setting the target of perioperative management. We have to understand the importance of the timing of initiation of nutritional therapy and selection of the route of administration.

### 2.3. Nutritional Management (Perioperative Management: Enhanced Recovery after Surgery)

Enhanced recovery after surgery (ERAS) aiming at postoperative early recovery by perioperative management was published in North Europe in 2005 [16]. This bundled 17 elements. For utilization for the gastrointestinal tract, there are four elements: removal of the preoperative fasting period, carbohydrate administration until immediately before surgery, early initiation of postoperative oral ingestion, and promotion of intestinal motility, and prevention of nausea and vomiting, and the use of the intestine as much as possible is mentioned. The ‘Clinical nutrition in surgery’ section of the ESPEN guidelines also specifies avoidance of long periods of preoperative fasting and re-establishment of oral feeding as early as possible after surgery to prevent prolongation of the preoperative fasting period [17]. Barlow et al. divided 121 patients undergoing upper gastrointestinal surgery (esophageal cancer, stomach cancer, and pancreatic cancer) into two groups. Nutrient administration was initiated within 12 h after jejunostomy in the early enteral nutrition (EN) group, whereas oral ingestion was initiated 7–10 days after surgery in the control group, and the influence on the clinical course was investigated [18]. Complications by wound infection and pneumonia were significantly less frequent in the EN group than in the control group, and the median length of hospital stay was also significantly shorter in the EN group (16 days) than in the control group (19 days), revealing that early enteral nutrition improved the clinical outcome.

Our hospital also reviewed postoperative management centering on intravenous nutrition in patients treated by distal gastrectomy and total gastrectomy, and it adopted early enteral nutrition initiated from postoperative day (POD) 1 by inserting an EN tube into the upper jejunum through the nose during surgery from 2001.

The usefulness of early initiation of enteral nutrition in the perioperative period for nutrition management is widely recognized as described above, but similar stress-reducing effects of the orally ingested amino acids, cystine and theanine, which induce an increase in GSH, have been reported [19,20].

## 3. Amino Acids Cystine and Theanine

Cystine is a dimer of the sulfur-containing amino acid cysteine formed by disulfide bonding (Figure 2), and it is contained in many food materials including meat [21]. It is abundant in keratin, which is protein of the hair and nails [22]. Cystine incorporated into cells is reduced by thioredoxin (TRX) and becomes cysteine [23]. Theanine is an umami ingredient contained in tea [24]. Theanine is decomposed to glutamic acid and ethylamine in the body (Figure 2) [25]. After entering cells, cysteine and glutamic acid are used together with glycine for the synthesis of GSH, one of substances with the strongest antioxidant effects in the body [26]. It has been reported that the addition of cystine to macrophages collected from human peripheral blood promotes intracellular GSH synthesis in a dose-dependent manner, and the addition of a theanine metabolite, glutamic acid, further increases GSH synthesis [27]. GSH also has important action for adjusting immune function, but a decrease in GSH under strong stress and inflammation was reported [28,29]. Accordingly, it is predicted that the promotion of GSH synthesis in a patient’s condition, reducing GSH, will strengthen the biological defense reaction.

### 3.1. Influence of Cystine/Theanine Administration on Exercise Load in Athletes

In athletes, the decline of immunity and poor physical condition occur after performing an exercise load stronger than normal, and the inhibitory effects of cystine/theanine have been reported. Murakami et al. set a practice menu applying a strong load of daily running distance longer than normal for 16 university long distance runners and divided the runners into a group ingesting 700 mg of cystine and 280 mg of theanine from 7 days before initiation of a training camp to 9 days after initiation (C/T group) and a group ingesting a placebo (control group) [30]. The degree of increase in the granulocyte count after the exercise load was significantly smaller in the C/T group on day 1 of camp, and the degree of decrease in the lymphocyte count was also significantly smaller in the C/T group. It was presumed that cystine/theanine inhibited the excessive inflammatory reaction and suppressed the decline of immunological function. In another report, Murakami et al. surveyed the effects of cystine/theanine on an exercise load stronger than normal in long-distance runners [31]. Fifteen runners were divided into a group ingesting 700 mg of cystine and 280 mg of theanine for 10 days before strong exercise (C/T group) and a group ingesting a placebo (control group). The rate of granulocytes significantly increased on day 10 of the camp in the control group, but no significant increase was noted in the C/T group. The rate of lymphocytes slightly decreased in the control group (*p* = 0.08), but no decrease was noted in the C/T group. CRP significantly increased in the control group, but no increase was noted in the C/T group. Based on this result, they also reported that cystine/theanine are effective for maintenance of the body condition by preventing strong exercise load-induced decline of immunity and excess inflammation. Similarly, Kawada et al. investigated the influence of cystine/theanine on immune function after a strong exercise load [32]. They divided 15 body builders into two groups. The C/T group orally ingested 700 mg of cystine and 280 mg of theanine and the control group orally ingested a placebo for 2 weeks. Both groups performed normal training in the first week (three times a week) and twofold training: six times/week in the second week and NK cell activity was measured before the initiation of training and on days 7 and 14. There was no change in NK cell activity between before training and day 7 in both groups, but when the frequency of exercise increased by two times, NK activity decreased to 69.2% of that before training in the control group, whereas it was 101.7% in the C/T group, demonstrating no decrease, being significantly different from that in the control group. This clarified that cystine/theanine restore NK cell activity reduced by the stress of performing exercise loads stronger than normal. They hypothesized that cystine/theanine administration maintained intracellular GSH, resulting in the absence of a decrease in NK cell activity.

Based on the above reports, it was concluded that cystine/theanine administration promoting GSH synthesis exhibits stress-reducing effects after exercise loads stronger than normal.

### 3.2. Application of Cystine/Theanine to Patients Undergoing Surgery

Cystine/theanine, which inhibit excess inflammation after strong exercise loads and prevent the decline of immunity, are expected to exhibit similar effects after surgery, causing excess stress for the body, although it is different from exercise load. The usefulness of cystine/theanine has been investigated in clinical cases and by animal experiments.

As shown Table 1, Miyachi et al. randomly allocated 33 patients who underwent distal gastrectomy for stomach cancer into two groups: 15 patients to the cystine/theanine administration group (C/T group) and 18 patients to the placebo administration group (control group). The C/T group received oral administration of 700 mg of cystine and 280 mg of theanine for 10 consecutive days (until POD5) from 5 days before surgery including the day of surgery, and the control group received placebo administration [20]. The other postoperative management was the same, and time-course blood sampling and REE measurement were performed. Blood interleukin (IL)-6 and CRP reached the maximum levels on the day following surgery, then decreased thereafter, but IL-6 was significantly lower in the C/T group on POD4, and CRP decreased to a significantly low level on POD7, demonstrating recovery. Regarding changes in body temperature, it remained lower in the C/T group than in the control group throughout the course and became significantly lower on POD5. The blood lymphocyte count decreased to approximately 30% of that before surgery on POD1 and recovered thereafter, but recovery was faster in the C/T group, and the difference was significant on POD7 (control group 57.9% vs. C/T group 71.2%). In contrast to the lymphocyte count, the granulocyte count increased to 150% of that before surgery on POD1, then decreased thereafter, and it was significantly lower in the C/T group on POD7 (control group 128.0% vs. C/T group 114.6%). REE increased after surgery. In the control group, it increased by 1.14 times of that before surgery on POD1, gradually decreased thereafter, and became 0.92 times on POD14. On the other hand, in the C/T group, the ratio was 0.99 on POD1, with no increase, and it was significantly different from that in the control group. It slightly increased thereafter, but the increase was inhibited to a maximum level of 1.06 times, observed only on POD5, and the ratio on POD14 was 0.88, being lower than that before surgery. It was clarified that cystine/theanine inhibit the postoperative increase in calorie consumption similar to early enteral nutrition. Changes in the parameters described above also clarified that cystine/theanine administration promotes early recovery after surgery, which led to clarification of the stress-reducing effects. The stress-reducing effects of 1 g of amino acids similar to those of postoperative early enteral nutrition may improve perioperative management in the future. In our hospital, a 10-day ingestion of cystine/theanine in the perioperative period (including the day of surgery) is incorporated into the clinical pathway of patients undergoing gastrointestinal tract surgery.

### 3.3. Supportive Data of Animal Experiment for Cystine/Theanine

The effects of cystine/theanine in clinical cases were described above. The usefulness has been also reported by animal experiments using a model simulating gastrointestinal surgery and acute inflammation (Table 2).

Shibakusa et al. laparotomized mice under anesthesia and prepared a simple laparotomy group in which the abdomen was closed after laparotomy without manipulation, and a small intestine manipulation model in which the small intestine was pulled out of the wound and abrasion with a cotton swab was applied twice to the entire small intestine, and then the wound was closed. The influence of cystine/theanine on postoperative recovery was investigated [33]. Using the small intestine manipulation model, 70 mg/kg of cystine/theanine (5:2 mixture) (C/T group) or vehicle (V group) was administered orally for 5 days before surgery, including the day of surgery. The blood IL-6 level markedly increased 2 h after surgery in the V group compared with that in the simple laparotomy group, but the increase was significantly inhibited in the C/T group compared with that in the V group. When GSH contained in the intestinal mucosa and small intestinal Peyer’s patch was measured, it was significantly lower in the V group than that in the simple laparotomy group, but in the C/T group it was maintained at a level similar to that in the simple laparotomy group, and a significant difference from that in the V group was noted. On comparison of the food intake and body weight in the four-postoperative-day period with those before surgery, these were significantly reduced in the V group, whereas the degrees of decrease were small in the C/T group, demonstrating a significant difference from those in the V group. Regarding changes in the body weight, it decreased from that before surgery in the V group, but increased in the C/T group. Regarding the quantity of spontaneous behavior considered to be one of the most important indices of recovery, behavior quantity was significantly suppressed in the V group compared with that in the simple laparotomy group, whereas in the C/T group behavior quantity recovered more rapidly than that in the V group, and a significant difference was noted on day 4. When energy consumption was measured 24 h after surgery using the same model, it decreased immediately after surgery (considered an influence of anesthesia), then increased in the V group, but the increase was significantly inhibited in the C/T group. Based on the above findings, it was confirmed that cystine/theanine administration reduced stress and accelerated recovery. A decrease in the blood and intramuscular GSH levels after surgery in humans has been reported. Demonstration of the effects of cystine/theanine inhibiting the decrease in GSH in the small intestine in the mouse manipulation model suggests that a decrease in GSH was similarly inhibited in clinical surgery [34]. Tanaka et al. analyzed the mechanism of the anti-inflammatory effects of cystine after stimulation with LPS using a human monocyte cell line, THP-1 cells [35]. They mentioned that after the addition of cystine to cells, cystine was reduced to cysteine in cells after stimulation with LPS and promotion of the production of an anti-inflammatory cytokine, IL-10, accompanying this reduction resulted in a decrease in production of an inflammatory cytokine, IL-6, i.e., anti-inflammatory action. As cysteine generated from cystine in cells may be utilized for GSH synthesis, the anti-inflammatory mechanism of cystine/theanine is outlined in Figure 3. The anti-inflammatory effects of cystine/theanine are exhibited at least through the following two actions: inhibition of stress-induced decrease in the GSH level and reduction of cystine to cysteine in cells. Furthermore, the reduction of the mortality by cystine/theanine administration in a peritonitis model using LPS, which induces severe inflammation, and an intestinal ischemia reperfusion model have been reported, and these studies support the efficacy of cystine/theanine [35,36]. These clinical studies and animal experiments suggest that cystine/theanine accelerate recovery from stress by preventing a decrease in the GSH level after surgery and inhibiting inflammatory reactions. Therefore, cystine/theanine may be termed stress-reducing amino acids.

## 4. Conclusions and Perspective

Metabolic changes centering on the endocrine, nervous, and immune systems occur after surgery, and energy consumption and protein catabolism increase. The metabolism returns to normal after the tuning-point in which recovery from the stress period starts. During hospital stay, management of the stress period over the turning point is necessary. To improve management, a reduction of the stress of surgery itself is one of the most important elements. In addition to this, perioperative management is important. ERAS proposed from North Europe is a bundle of many elements to reduce stress, its effects have been demonstrated at clinical sites, and its use is spreading worldwide [37,38]. ERAS recommends avoiding fasting before surgery as much as possible and early use of the gastrointestinal tract. We also noted the stress-reducing effects of early enteral nutrition in patients treated by gastrectomy. It was also mentioned in this review that, based on animal experiments and clinical studies, the amino acids cystine/theanine suppress the decrease in GSH in cells and exhibit the surgical stress-reducing effects through anti-inflammatory and immune adjustment actions. The dose in humans is 700 mg of cystine and 280 mg of theanine; being low, compliance with ingestion is high, and their use is versatile and highly safe [39]. Therefore, the goal of perioperative management, especially the goal of nutritional therapy, should not be set to supplement lost energy and protein, but to early recovery to a state in which administered energy is normally metabolized, i.e., how to reduce stress should be set as the first target (Figure 4). By achieving this, it may result in the inhibition of postoperative complications and a shortening of hospital stay.

The possibility of other applications of cystine/theanine has also been investigated. Reduction of the mortality from irradiation and effects as a radiation protective agent, such as a protection from destruction of the small intestinal mucosa, have been clarified by rat experiments [40]. In addition, clinical studies and animal experiments reported an effect against adverse events of anticancer drugs such as S-1 (combination of tegafur, gimeracil, and oteracil), oxaliplatin, and capecitabine (Table 3 and Table 4) [41,42,43,44]. Tsuchiya et al. investigated the effects of reducing the side effects of cystine/theanine intake in patients with colorectal cancer and gastric cancer who received S-1 as adjuvant chemotherapy [20]. As a result, the completion rate of treatment was significantly higher in the C/T group than in the control group. As for side effects, the frequency of diarrhea was significantly reduced. When the effects of 5-FU on diarrhea were examined using mice, cystine/theanine administration significantly suppressed diarrhea and also suppressed the decrease in food intake and body weight [45]. Histological analysis showed that 5-FU-induced villus destruction was suppressed by cystine/theanine administration and that the decrease in GSH level in the mucosa was also suppressed. From these results, it is reported that preventing the decrease in GSH is considered to be one of the mechanisms for preventing diarrhea by cystine/theanine intake. In addition, oxaliplatin, which is a key drug for colorectal cancer chemotherapy, causes peripheral neuropathy as a side effect, leading to a decrease in the QOL of patients. Kobayashi et al. investigated the effects of cystine/theanine intake on peripheral neuropathy in patients with colon cancer who received mFOLFOX6 therapy including oxaliplatin as a postoperative adjuvant chemotherapy [43]. As a result, peripheral neuropathy was significantly suppressed in the C/T group. In animal experiments, it has been reported that cystine/theanine administration significantly suppresses peripheral neuropathy caused by oxaliplatin administration [42]. In addition, multicenter collaborative studies have shown that cystine/theanine intake suppresses hand–foot syndrome, a major side effect of capecitabine in many cancers [41]. Accordingly, cystine/theanine are expected to be effective not only in the perioperative period but also against many pathologies accompanying inflammation and oxidative stress in clinical cases. Research results identifying them as stress-reducing amino acids in the future are awaited.

## Figures and Tables

**Figure 1 nutrients-14-00129-f001:**
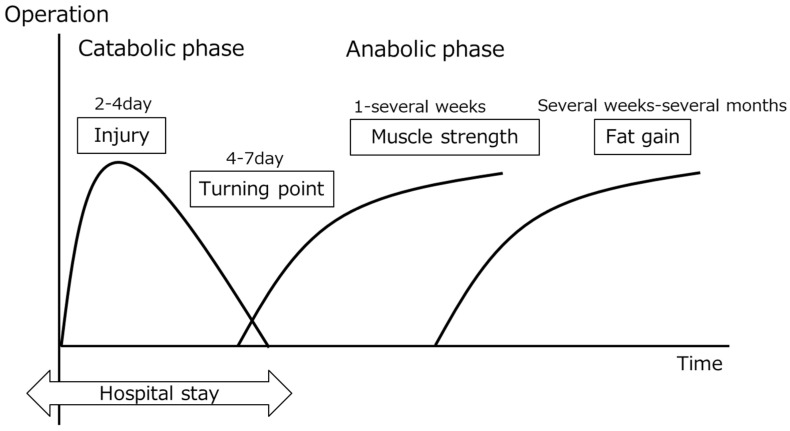
Metabolic changes after surgery. The metabolic state must be understood in nutritional therapy during a hospital stay.

**Figure 2 nutrients-14-00129-f002:**
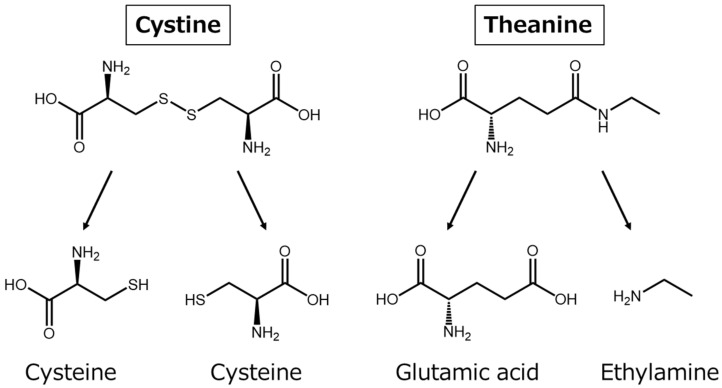
Chemical structure of cystine and theanine and metabolism in the body.

**Figure 3 nutrients-14-00129-f003:**
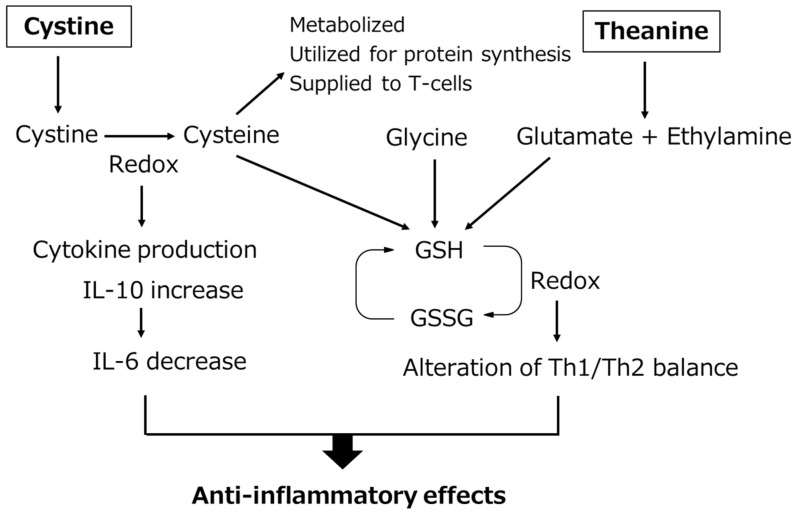
Working hypothesis for the anti-inflammatory effects of cystine/theanine. Cystine/theanine have anti-inflammatory effects both through the maintenance of GSH levels and the promotion of IL-10 production. GSH: glutathione, GSSG: glutathione disulfide, IL-10: interleukin-10, IL-6: interleukin-6, Th1/Th2: type 1 helper T cells/type 2 helper T cells.

**Figure 4 nutrients-14-00129-f004:**
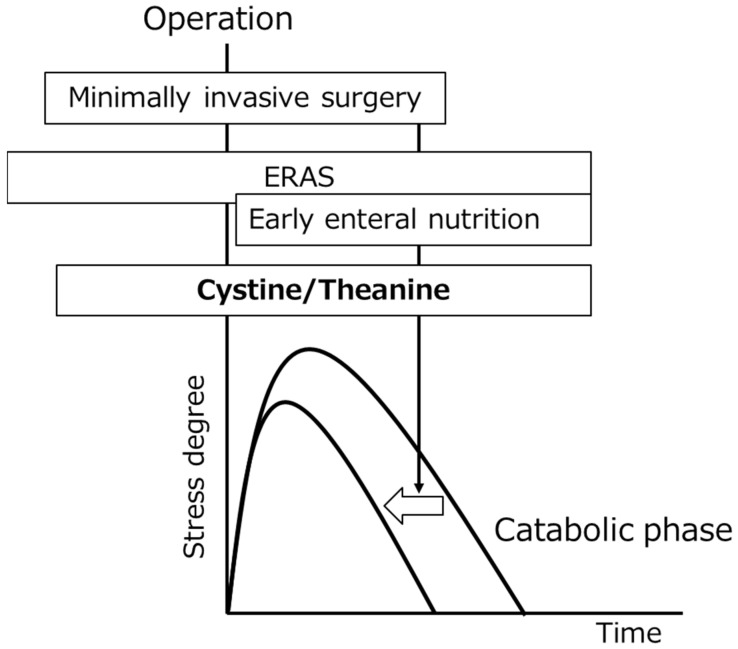
Perioperative management. To shorten the catabolic phase, minimally invasive surgery and nutritional therapy, including postoperative early enteral feeding (an element of ERAS) and oral intake of cystine/theanine, are effective. ERAS: enhanced recovery after surgery.

**Table 1 nutrients-14-00129-t001:** Clinical study of cystine/theanine for perioperative management.

Author and Year	Study Design	Number of Patients	Study Results	Conclusions
Miyachi, et al. (2013) [20]	RCT	C/T group: 15 placebo group: 18	Significant decrease in C/T group: IL-6, CRP, body temperature and REE.	C/T reduces excessive inflammation after surgery and enhances recovery.

RCT: randomized controlled trial, C/T: cystine/theanine, IL-6: interleukin-6, CRP: C-reactive protein, REE: resting energy expenditure.

**Table 2 nutrients-14-00129-t002:** Basic research of cystine/theanine for anti-inflammatory effects.

Author and Year	Study Model in Mice/Cells	Study Results	Conclusions
Shibakusa, et al. (2012) [33]	Small intestine manipulation model in mice	Significant decrease in C/T group: IL-6. Significant increase in C/T group: GSH level, food intake and locomotor activity.	C/T reduces excessive inflammation after surgery and enhances recovery.
Tanaka, et al. (2015) [35]	LPS-induced sepsis model in mice/LPS-treated THP-1 cells	Significant decrease in C(/T) group: IL-6. Significant increase in C(/T) group: survival rate and IL-10 production.	C(/T) reduces excessive inflammation after LPS treatment through enhancing IL-10 production and recovers survival rate.
Miyakuni, et al. (2018) [36]	Intestinal ischemia reperfusion model in mice	Rapidly decrease in C/T group: IL-6 and TNF-α. Significant increase in C/T group: survival rate.	C/T reduces excessive inflammation after an intestinal ischemia reperfusion and recovers survival rate.

LPS: lipopolysaccharide, THP-1: a human monocytic leukemia cell line, C/T: cystine/theanine, C(/T): cystine and/or theanine, IL-6: interleukin-6, GSH: glutathione, IL-10: interleukin-10, TNF-α: tumor necrosis factor-alpha.

**Table 3 nutrients-14-00129-t003:** Clinical study of cystine/theanine for adverse events of anticancer drugs.

Author and Year	Study Design	Number of Patients	Study Results	Conclusions
Tsuchiya, et al. (2016) [44]	RCT	C/T group: 32 Placebo group: 31	C/T improved the completion rate, alleviated the adverse events, especially diarrhea in colon and gastric cancer patients.	C/T reduces adverse event of S-1 adjuvant chemotherapy.
Hamaguchi, et al. (2019) [41]	RCT	C/T group: 52 Placebo group: 48	C/T reduced diarrhea and hand foot syndrome in colon cancer patients, but it was not significantly different	C/T has a possibility to reduce diarrhea and hand foot syndrome of capecitabine.
Kobayashi, et al. (2020) [43]	RCT	C/T group: 14 Control group: 14	C/T reduced neuropathy grading score during mFOLFOX chemotherapy in colon cancer patients.	C/T has a protective effect against peripheral neuropathy induced by oxaliplatin.

RCT: randomized controlled trial, C/T: cystine/theanine, S-1: combination of tegafur, gimeracil and oteracil, mFOLFOX: leucovorin, 5-fluorouracil and oxaliplatin.

**Table 4 nutrients-14-00129-t004:** Basic research of cystine/theanine for adverse events of anticancer drugs.

Author and Year	Study Model in Rats/Mice	Study Results	Conclusions
Kawashiri, et al. (2020) [42]	Oxaliplatin-induced peripheral neuropathy model in rats	Significant increase in C/T group: GSH level. Significant suppress in C/T group: axonal degeneration.	C/T enhances GSH level and suppresses peripheral neuropathy induced by oxaliplatin.
Yoneda, et al. (2021) [45]	5-FU-induced diarrhea model in mice	Significant improvement in C/T group: GSH level, villus destruction, diarrhea, food intake and body weight.	C/T enhances GSH level and suppresses diarrhea induced by 5-FU.

C/T: cystine/theanine, 5-FU: 5-fluorouracil, GSH: glutathione.

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
