# Peer review of "Cystine and Theanine as Stress-Reducing Amino Acids—Perioperative Use for Early Recovery after Surgical Stress"

_nutrients, 2021, doi:10.3390/nu14010129_

Round 1

Reviewer 1 Report

Major

Introduction - Section 2.2

“When the rate of the blood lymphocyte count relative to that before surgery at 2 weeks after surgery was compared, recovery was up to 84.3% in the control group, but it was 102.3%, the preoperative level, in the EN group, demonstrating a significant difference between the groups [19].”

→ Authors may add a short conclusive sentence to clarify the point about the lymphocyte count. Everybody might not be comfortable with the expected normal values.

End of introduction

“The usefulness of early initiation of enteral nutrition in the perioperative period for nutrition management is widely recognized as described above, but similar stress-reduc-ing effects of orally ingested amino acids, cystine and theanine, which induce an increase in GSH, have been reported [20].”

The authors cited only one reference. They might elaborate a bit more with additional references. The article cited is a review, whereas we can find in the literature original articles including a randomized clinical trial conducted by one of the authors by the way.

About CRP :

The authors described very well the work done around the measurement of CRP. Why didn't they elaborate more on the work done with PCT, which has shorter kinetics?

Section 3.2

Changes in the parameters described above also clarified that cystine/theanine administration promotes early recovery after surgery, which led to clarification of the stress-reducing effects. The stress-reducing effects of 1 g of amino acids similar to those of postoperative early enteral nutrition may improve perioperative management in the future. In our hospital, 10-day ingestion of cys-tine/theanine in the perioperative period (including the day of surgery) is incorporated into the clinical pathway of patients undergoing gastrointestinal tract surgery.

 What are the authors conclusions ? They described the decreased inflammation parameters but we would know if they could demonstrate data of a reduction in the length of stay, shortened functional recovery, etc. ?

 What about giving ORAL IMPACT, which is also prescribed for the same purpose of preoperative immunonutrition ?

Conclusion

The conclusions should concisely restate the points discussed and detailed in the review. Unfortunately, the authors did not show strong evidence to support conclusions of shortened clinical recovery or reduction in postoperative complication rates related to cystin/theanine consumption.

Reviewer 2 Report

This is an review article on the role of cystine and theanine as stress reducing aminoacids in the aspect of perioperative use for early recovery after surgical stress. The topic is interesting and clinically relevant. Some issues should be resolved:

  1. The paragraph "Perioperative management" should be divided into parts including studies conducted on humans and animals. In the current for, it is not clear.
  2. The paragraph "Amino acids cystine and theanine" should contain more information regarding characteristics of cystine and theanine. It should be divided into two parts concerning cystine and theanine separately. Figures presenting the chemical structure of cystine and theanine should be added. I recommend to present this paragraph following the introduction.
  3. There is an extensive paragraph "Influence of cystine/theanine administration on exercise load in athletes", but there is no information regarding oral administration of cystine and theanine attenuating the adverse events of adjuvant chemotherapy in gastrointestinal cancer patients or in colorectal cancer patients undergoing capecitabine-based adjuvant chemotherapy after surgery. The clinical use of cystine and theanine in patients, especially undergoing surgery, is more important for clinicians than the use in athletes. In the current form, there is a large disproportion between these applications. Regarding this clinical use, only one sentence in the paragraph "Conclusions and perspective" is presented.
  4. I recommend to add table summarizing the most important presented studies including the author's name, study design, number of patients, results/conclusions.

Round 2

Reviewer 1 Report

The authors have well addressed my comments. I congratulate them for their work.

Reviewer 2 Report

The authors have improved their work taking into account most of my suggestions. I recommend to summarize in the table a larger number of presented in this review studies, not only clinical trial. Currently only the one clinical study is included in the table. The larger table divided into several parts presenting different presented in this article studies should be considered.
